# Cytotoxic Effects of the Dual ErbB Tyrosine Kinase Inhibitor, Lapatinib, on Walker 256 Rat Breast Tumour and IEC-6 Rat Normal Small Intestinal Cell Lines

**DOI:** 10.3390/biomedicines8010002

**Published:** 2019-12-30

**Authors:** Wan Nor I′zzah Wan Mohamad Zain, Joanne Bowen, Emma Bateman, Dorothy Keefe

**Affiliations:** 1Faculty of Medicine, Universiti Teknologi MARA, Sungai Buloh Campus, Jalan Hospital, Sungai Buloh 47000, Selangor, Malaysia; 2Adelaide Medical School, University of Adelaide, Adelaide, SA 5005, Australia; joanne.bowen@adelaide.edu.au (J.B.); emma.bateman@adelaide.edu.au (E.B.); 3Cancer Centre, Royal Adelaide Hospital, Port Road, Adelaide, SA 5000, Australia; Dorothy.Keefe@canceraustralia.gov.au

**Keywords:** lapatinib, ErbB1/ErbB2 TKI, Walker 256, IEC-6, diarrhoea

## Abstract

Lapatinib is an orally administered, dual ErbB1/ErbB2 tyrosine kinase inhibitor (TKI). It is effective in ErbB2 + ve breast cancer treatment. However, lapatinib is associated with diarrhoea with an incidence of 47–75%. The mechanism of ErbB1 TKI-induced diarrhoea remains unclear. ErbB1 or epidermal growth factor receptor (EGFR) is expressed in gastrointestinal mucosa whereby the primary site for drug absorption is intestine. Thus, administration of ErbB1 oral TKI may disrupt gut homeostasis, leading to diarrhoea. Nevertheless, further investigations are required. We observed that lapatinib inhibited 50% Walker 256 breast tumour cells and IEC-6 small intestinal cell growth. Higher percentage of necrosis was observed in lapatinib-treated Walker 256. Lapatinib-treated IEC-6 showed higher percentage of late apoptosis. Only ErbB2 mRNA was detected in Walker 256 but both ErbB1 and ErbB2 mRNAs were detected in IEC-6, yet both protein staining were detected in both cells. Lapatinib exhibited cytotoxic properties on ErbB1/ErbB2 expressing cell lines, with intestinal cells being more sensitive to lapatinib compared to tumour cells. Lapatinib induced necrosis in tumour cells, while inducing late apoptosis in intestinal cells may explain lapatinib-induced diarrhoea in patients administered with the drug which could be due to apoptosis of intestinal epithelial cells leading to barrier disruption and consequently diarrhoea.

## 1. Introduction

ErbB1 and ErbB2 play significant roles in ligand-activated signalling pathways that regulate cell proliferation and cell death [1]. Overexpression of ErbB1 and/or ErbB2 has been implicated in the development and progression of various cancer types such as head and neck, lung, pancreas, bladder, breast, colon, ovary and bladder [2]. ErbB2 is the most common heterodimerisation partner of ErbB1 in which ErbB2 potentiates ErbB1 signalling via an in trans mechanism by enhancing the binding affinity of the ErbB1 ligand, EGF, reducing its degradation and predisposing the receptor to recycling [3,4]. ErbB1-specific inhibitors have been proven to reduce ErbB2-signalling and growth of breast cancer cells that express high levels of ErbB2 [5,6]. ErbB2 has also been reported to play an important role in the oncogenic activity of ErbB1 [3]. As such, combined inhibition of both ErbB1 and ErbB2 may be more efficacious than targeting either one of them. Thus, these proteins show potential as targets for anticancer drugs [7].

Lapatinib is an orally administered small molecule dual tyrosine kinase inhibitor of ErbB1 and ErbB2 and has been approved for the treatment of metastatic breast cancer patients [7,8,9]. Structural and biochemical studies have indicated that lapatinib in vitro is a potent inhibitor of the tyrosine kinase activity of ErbB1 and ErbB2, by which lapatinib binds to an inactive-like conformation of ErbB1 and has a slow dissociation rate (half-life ≥ 300 min) from its target receptor in vitro and in vivo [10] which may result in prolonged inhibition of tyrosine kinase activity of tumour cells. Lapatinib reversibly inhibits ErbB1 and ErbB2 tyrosine kinases leading to inhibition of Ras/Raf MAPK and PI3K/Akt pathways, leading to an increase in apoptosis and a decrease in cellular proliferation [11].

Lapatinib cytotoxic activity has been tested on several cancer cell lines such as gastric, head and neck, lung, epidermal, breast and others and it has been proven that lapatinib inhibits the cancer cell growth at low concentrations within the range of 0.01 to 9.8 µM [12,13]. The studies also show that the cytotoxic activity of lapatinib on the cancer cell lines is due to the overexpression of ErbB1 and/or ErbB2 or varying levels of both ErbB1 and ErbB2 expression [12,13]. Among all the cancer cell lines tested, lapatinib has shown to have the highest cytotoxic effect on the breast cancer cell lines (UACC182: 0.01 µM, SUM190: 0.018 µM, BT474: 0.022 µM, SK-BR-3: 0.037 µM, SUM225: 0.083 µM and MDA-MB-453: 3.9 µM) and this correlated to higher ErbB2 expression (108 to1161 ng/mg) [12,13]. However, there were also breast cancer cell lines which showed high lapatinib cytotoxic effect (MDA-MB-175: 0.012 µM, EFM-19: 4.6 µM, MCF-7: 7.7 µM and MDA-MB-435: 8.5 µM) but expressed a moderate or low level of both ErbB1 and ErbB2 (2.2 to 20 ng/mg) [12,13]. There were also breast cancer cell lines with high lapatinib cytotoxic effect (MDA-MB-468: 4.7 µM and BT20: 9.8 µM) that correlated with higher ErbB1 expression (295 to 908 ng/mg). Thus, much remains to be learned about the molecular mechanisms that determine lapatinib sensitivity in cancer cell lines.

This study is conducted to determine in vitro toxicity of lapatinib on tumour and intestinal cell lines which may explain the significant association between lapatinib and gastrointestinal toxicities, particularly diarrhoea [14,15,16]. Lapatinib is effective on human tumour cell lines, however, no study has been conducted to evaluate the effect of lapatinib on normal intestinal cell lines. As far as concerned, lapatinib also has not yet been tested on any rat cell line including rat breast tumour and intestinal cell lines. Two cell lines, Walker 256 and IEC-6, were used in this study. Walker 256 is a rat breast carcinoma syngeneic to Wistar rats and commonly used to induce secondary brain tumours [17] and bone metastases [18]. It has also been used to develop cachectic tumour-bearing rat model to study metabolic changes characteristic of cancer cachexia [19,20]. Lapatinib is known to be effective in inhibiting breast cancer metastasis; hence Walker 256 breast tumour cell line was used in this study. Clinically, while inhibiting cancer metastasis, at the same time, lapatinib also has a significant effect on the gastrointestinal tract; thus IEC-6 was used in this study. IEC-6 is a non-transformed intestinal cell line derived from rat jejunal crypt cells. It is a homogeneous population of epithelial-like cells commonly used as a model to elucidate the mechanism of intestinal epithelial cell differentiation, growth and wound healing [21]. Use of these cell lines allows for further investigation of the underlying mechanisms on lapatinib efficacy on breast cancer cells, as well as its toxicity on the normal intestinal cells. Understanding the effect of lapatinib on Walker 256 rat breast tumour and IEC-6 rat normal intestinal cells would increase the potential of the cell lines to be used in developing an appropriate rat tumour model to study tyrosine kinase inhibitor-induced gastrointestinal toxicity, in particular lapatinib-induced gastrointestinal toxicity to reflect breast cancer patient setting receiving lapatinib treatment.

## 2. Experimental Section

### 2.1. Reagents and Antibodies

The growth media: RPMI-1640 and Dulbecco′s modified Eagle medium (DMEM), 1× phosphate buffered saline (PBS), L-glutamine, penicillin-gentamicin with fungizone and trypsin were supplied by Gibco^®^, Life Technologies Pty Ltd., Mulgrave, VIC, Australia. Dimethyl sulfoxide (DMSO), ethylenediaminetetraacetic acid (EDTA) solution and 0.4% trypan blue were purchased from Sigma Aldrich, Castle Hill, NSW, Australia. Foetal bovine serum (FBS) was from Bovogen PtyLtd., Melbourne, VIC, Australia. 2,3-*bis*-(2-methoxy-4-nitro-5-sulfophenyl)-2H-tetrazolium-5-carboxanilide solution (XTT) was purchased from Roche Diagnostics GmbH, Mannheim, Germany. Fluorescein isothiocyanate (FITC) Annexin V Apoptosis Detection Kit I, which consists of 10× Annexin V binding buffer, FITC Annexin V and propidium iodide (PI) staining solutions was obtained from BD Pharmingen, San Diego, CA, USA. Rabbit polyclonal anti-ErbB1 (0.2 mg/mL) and anti-ErbB2 (0.2 mg/mL) were supplied by Abcam, UK. Fluorophore-labelled donkey anti-rabbit IgG (H + L) secondary antibody Alexa Fluor^®®^ 568 (2 mg/mL) was purchased from Invitrogen, Carlsbad, CA, USA. Normal rabbit IgG control (polyclonal rabbit IgG) was obtained from R&D Systems, Noble Park North, VIC, Australia while 4′,6-diamidino-2-phenylindole dihydrochloride (DAPI) (1 mg) was from Sigma Aldrich, Castle Hill, NSW, Australia. Lapatinib (GlaxoSmithKline (GSK), Melbourne, VIC, Australia) was dissolved in 100% DMSO to 10 mM. Drug solutions for treatment were diluted with serum-free medium from the stock solution.

### 2.2. Cell Culture

The Walker 256 rat breast carcinoma cell line was obtained from the Cell Resource Centre for Medical Research at Tohoku University, Japan. The IEC-6 rat small intestinal cell line was obtained from the American Type Culture Collection (ATCC), USA. Walker 256 cells are derived from female Wistar rat breast carcinoma while IEC-6 cells are derived from adult rat jejunum. Assays using these cell lines were carried out between passages 2 and 10. All of the cell lines had tested negative for mycoplasma contamination by Dr Nicholas Eyre from Hepatitis C Virus Research Laboratory, School of Molecular and Biomedical Science, University of Adelaide, Australia. Walker 256 cells were grown in RPMI-1640 (supplemented with 10% FBS and 1% penicillin-gentamicin with fungizone). IEC-6 was grown in DMEM supplemented with 10% FBS, 1% penicillin-gentamicin with fungizone and 2mM L-glutamine. All of the cells were cultured in 75-cm^2^ flasks in a 37 °C incubator with 5% CO_2_. In order to subculture, the cells were divided and the culture medium replaced with fresh medium as follows; first, the old medium was removed, and then the cells were rinsed briefly with PBS. Walker 256 cells were subcultured with 2 mL of 0.02% EDTA in PBS while IEC-6 cells were subcultured with 2–3 mL of 0.01% trypsin in PBS. Flasks were incubated at 37 °C and 5% CO_2_ for 3–5 min. After the cells had detached from the lower surface of the flask, 20 mL of medium was added to the flask (to stop the enzymatic digestion) and the culture was divided in two parts. One part was then transferred to a new flask.

### 2.3. Cell Viability

Walker 256 and IEC-6 viability were evaluated by using XTT assay. One hundred microliters of suspension containing 5 × 10^3^ cells were seeded in each well of a 96-well microtiter flat-bottom plate (Becton Dickinson, Bedford, MA, USA). The plate was then incubated for 24 h at 37 °C with 5% CO_2_. Lapatinib stock (10 mM) was diluted with serum-free medium to a concentration series (1–10 µM). After 24 h incubation, the old medium was replaced and the cells were treated with lapatinib at the dose of 1–10 µM and incubated at 37 °C with CO_2_ for 48 h. An equivalent serial dilution of DMSO was used as the control treatment. After the incubation period, the old media was replaced with 100 µL fresh media and 50 µL of XTT solution (composed of 5 mL XTT labelling reagent and 100 µL of electron coupling) was added to each well. The microtiter plate was then incubated again for 6 h at 37 °C with 5% CO_2_. Then, the cell viability was measured using an ELISA reader (Bio-Rad 550, San Diego, CA, USA) at 490 nm. A graph of percentage of cell viability versus concentration of lapatinib was then plotted and the IC_50_ (a dose that inhibited 50% cell growth) was determined from the graph. The experiment was repeated until the final IC_50_ value of lapatinib was confirmed. The final concentration of lapatinib that had been determined to inhibit 50% cell growth was used in the following experiments and cells were incubated for 6, 24 and 48 h to evaluate the cell growth at different time points.

### 2.4. Flow Cytometric Analysis of Cell Death Stages

This experiment was carried out to evaluate cell apoptosis. Each cell line was grown in a sterile tissue culture dish plate (Becton Dickinson, Bedford, MA, USA) in a final volume of 10 mL culture medium per dish that contained 5 × 10^4^ cells/mL and was incubated for 24 h at 37 °C with 5% CO_2_. Cells were treated with lapatinib and incubated for 6, 24 and 48 h. After the incubation period with lapatinib, the old medium potentially containing dead cells was collected in a 50 mL falcon tube. The remaining cells were then dislodged and the cell solution was added to the 50 mL falcon tube. The tubes were spun at 1400 rpm for 5 min. Supernatant was removed and the cells were washed with cold PBS and spun again for 2 min. The supernatant was then removed and the cell pellet was resuspended in 100 µL of 1× Annexin V binding buffer and transferred to a 5 mL FACS tube. Five microlitres of FITC Annexin-V and propidium iodide (PI) were added to the tubes. Cells were gently vortexed and incubated for 15 min at room temperature in the dark. Four hundred microliters of 1× Annexin V binding buffer was then added to each tube. Three tubes of untreated cells were prepared as follows - unstained cells, cells stained with FITC Annexin V (no PI) and cells stained with PI (no FITC Annexin V), in order to set up compensation and quadrants for flow cytometry analysis. Samples were kept on ice and analysed by a flow cytometry machine (FACSCalibur, BD Biosciences, San Jose, CA, USA) within 1 h. At least 10,000 cells were examined in the gated region used for calculation. Dual parameter cytometric data were analysed using CellQuest software from BD Biosciences. Viable cells were indicated as FITC Annexin V and PI negative, whereas FITC Annexin V negative and PI positive staining indicated necrosis, FITC Annexin V positive and PI negative staining indicated early apoptosis, and cells that were FITC Annexin V and PI positive were considered to be in late apoptosis.

### 2.5. RNA Extraction, Reverse Transcription and Real-Time PCR

Cell lines were subcultured into a sterile tissue culture dish plate (Becton Dickinson, Bedford, MA, USA) in a final volume of 10 mL culture medium per well that contained 5 × 10^4^ cells/mL and was incubated for 24 h at 37 °C with 5% CO_2_. The cells were then treated with lapatinib at a dose that inhibited 50% cell growth (IC_50_). Cultures were maintained at 37 °C with CO_2_ for 6, 24 and 48 h. After the incubation period, the cells were dislodged using the subculture procedure and total RNA was isolated from the cells using Trizol and followed by mRNA purification using the Nucleospin^®®^ mRNA purification RNA II kit (Macherey–Nagel, Düren, Germany) following the manufacturer′s protocol. Up to 1 μg of RNA was reverse transcribed using iScript cDNA Synthesis Kit (Bio-Rad, Hercules, CA, USA) according to the manufacturer′s protocol. cDNA was quantified and diluted to a working concentration of 100 ng/μl. Primers for genes of interest (shown in Table 1), were designed using web-based primer design program, PRIMER 3 version 4 [22]. Amplified transcripts were detected by SYBR Green (Quantitect, Qiagen Pty Ltd., Chadstone Centre, VIC, Australia) in a Rotor-Gene Q Series Rotary Cycler (Qiagen Pty Ltd., Chadstone Centre, VIC, Australia). All reactions were completed in triplicate, and amplification was followed by a melt curve analysis to confirm product specificity. Fold change in *ErbB1* and *ErbB2* mRNA expression was calculated using Delta CT (2^−ΔCt^) method. The experimental threshold (Ct) values were calculated manually by converting the Ct values into relative quantities relative to two housekeeping genes which are *UBC* and *B2M*.

### 2.6. Immunofluorescence Staining of ErbB1 and ErbB2

Cells (5 × 10^4^ cells/mL) were seeded in 8-well chamber slide (Millicell^®®^ EZ Slide, Burlington, MA, USA) which were coated with poly-*L*-lysine (Sigma Aldrich, Castle Hill, NSW, Australia) (1:10) prior to cell seeding. Cells were then incubated for 48 h to allow cell adherence. Then, cells were treated with lapatinib and incubated for 6, 24 and 48 h. After the incubation period, the medium was carefully aspirated. Cells were then washed with PBS for 1 min. Next, PBS was aspirated prior to cell fixation in 4% paraformaldehyde (Sigma Aldrich, Castle Hill, NSW, Australia) for 20 min at room temperature. The fixative solution was aspirated prior to 2 × 5 min washing in PBS and following that, cells were permeabilised with 0.5% Triton-X solution (Sigma Aldrich, Castle Hill, NSW, Australia) for 10 min at room temperature. Next, cells were rinsed with PBS for 3 × 5 min and were blocked with 1% bovine serum albumin (BSA) (Sigma Aldrich, Castle Hill, NSW Australia) in PBS for 1 h at room temperature. After the incubation h, the blocking solution was aspirated and cells were incubated with primary antibodies diluted in 1% BSA (ErbB1: 5 µg/mL, ErbB2: 5 µg/mL) for 18 h at 4 °C. Then, the primary antibodies were aspirated and cells were washed with PBS for 2 × 3 min. Following that, cells were incubated with fluorophore-labelled secondary antibodies diluted in 1% BSA (5 µg/mL) for 1 h at room temperature, protected from light. Following washing with PBS for 2 × 3 min, cells were incubated with DAPI diluted in PBS (1 µg/mL) for 15 min at room temperature, protected from light. The cells were washed again in PBS (2 × 3 min), then the chamber was removed from the 8-well slide and the slide was coverslipped with an aqueous mounting media, Fluoroshield (Sigma Aldrich, Castle Hill, NSW, Australia). The coverslip was then sealed with nail polish and left to dry at room temperature for 20 min prior to storage at 4 °C. Slides were viewed using a confocal microscope (Leica TCS SP5, Leica Microsystems, Mannheim, Germany) within 48 h. Cells treated with rabbit IgG (5 µg/mL) or BSA (1%) only were used as controls for comparison. To evaluate the expression of ErbB1 and ErbB2 (positive fluorescein) in control untreated and lapatinib-treated Walker 256 and IEC-6 cells, immunofluorescent images from six random fields were analysed using ImageJ software (http://imagej.nih.gov/ij/; National Institutes of Health, Bethesda, MD, USA). Corrected total cell fluorescence (CTCF) was calculated by subtracting area of selected cell multiplies with mean fluorescence of background readings from the integrated density.

### 2.7. Statistical Analysis

Data were presented as mean ± S.E.M. All analyses were performed using GraphPad Prism version 7.04. Statistical analysis of ErbB2 mRNA expression results in Walker 256 cells were carried out using two-way ANOVA with Sidak’s multiple comparisons test to identify the differences between control untreated samples and treated samples at different incubation time. One-way ANOVA with Tukey’s multiple comparisons test was used to analyse ErbB1 and ErbB2 mRNA expression in IEC-6 cells. FACS results and ErbB1 and ErbB2 immunofluorescence staining intensities were analysed using two-way ANOVA with Tukey’s multiple comparisons test to compare means of each group at different incubation time. Statistical significance was accepted as *p* < 0.05.

## 3. Results

### 3.1. Lapatinib Inhibited Cell Proliferation in Walker 256 and IEC-6

Walker 256 and IEC-6 were treated with lapatinib at a series of concentrations (1–10 µM) to determine the lapatinib dosage that could inhibit 50% cell growth (Figure 1a). Lapatinib was found to inhibit 50% of Walker 256 rat breast tumour cell growth at 8.40 ± 0.83 µM, and at 3.00 ± 0.96 µM in the IEC-6 rat jejunum cell line. Experiments were also carried out with DMSO (lapatinib vehicle), which was assayed in a series of concentrations equivalent to the concentration of lapatinib treatment. DMSO did not cause 50% cell inhibition (Figure 1b) at any of the concentrations, which signifies that the vehicle did not influence lapatinib cytotoxic effect on both cell lines.

### 3.2. Mechanism of Cell Death Induced by Lapatinib

As indicated in the results above, lapatinib was shown to inhibit cell death in both Walker 256 and IEC-6 cells. Thus, flow cytometry was carried out to evaluate the mechanism of cell death induced by lapatinib. Percentage of viable, early apoptotic, late apoptotic and necrotic cells in Walker 256 and IEC-6, after treatment with lapatinib at different incubation time were presented in Figure 2a–c (Walker 256) and Figure 2d–f (IEC-6). At 6 h, lapatinib-treated samples showed a significantly lower number of viable cells (58.99 ± 3.21%) (*p* < 0.0001) and higher numbers of early apoptotic cells (24.71 ± 1.39%) (*p* < 0.0001), compared to control untreated (viable cells: 79.97 ± 0.99%, early apoptotic cells: 7.30 ± 2.51%) (Figure 2a), as determined by flow cytometry. However, lapatinib-treated samples did not show any difference in the percentage of viable, early apoptotic, late apoptotic and necrotic cells at 24 h incubation (Figure 2b) compared to control untreated samples (*p* > 0.05), while at 48 h incubation, lapatinib-treated samples were shown to have a lower percentage of viable cells (50.70 ± 7.27%) (*p* < 0.05) and higher percentage of necrotic cells (37.91 ± 7.08%) (*p* < 0.01), compared to control untreated samples (viable cells: 71.93 ± 6.71%, necrotic cells: 11.86 ± 5.62%) (Figure 2c).

As for IEC-6, the results did not show any significant differences in cell viability at 6 h incubation (*p* > 0.05) (Figure 2d). However, lapatinib-treated samples at 24 h incubation showed a lower percentage of viable cells (27.72 ± 9.59%) (*p* < 0.05) and a higher percentage of late apoptotic cells (53.56 ± 15.37%) (*p* < 0.01) compared to control untreated samples (viable cells: 65.00 ± 9.70%, late apoptotic cells: 12.91 ± 4.70%) (Figure 2e). Similarly, at 48 h incubation lapatinib-treated samples showed a lower percentage of viable cells (25.68 ± 10.78%) (*p* < 0.05) and a higher percentage of late apoptotic cells (56.82 ± 11.53%) (*p* < 0.05) compared to the control untreated samples that exhibited 65.83 ± 13.11% alive cells and 22.70 ± 12.81% late apoptotic cells (Figure 2f).

### 3.3. ErbB1 and ErbB2 mRNA Expression

ErbB1 was unable to be detected in Walker 256 cells because of the low expression (no data shown in Figure 3 as ErbB1 was unable to be detcted). ErbB2 mRNA expression in lapatinib-treated Walker 256 cells was not significantly different compared to control untreated cells at 6, 24 and 48 h incubation (*p* > 0.05) (Figure 3a). As for IEC-6, ErbB1 mRNA expressions increased as incubation time increased in both control untreated and lapatinib-treated cells. However, the results were not significantly different between lapatinib-treated cells and control untreated cells (Figure 3b) (*p* > 0.05). Figure 3c illustrates ErbB2 mRNA expression in IEC-6 cells in which control untreated cells showed increasing ErbB2 mRNA expressions with the increase of incubation h, while lapatinib-treated cells indicated decreasing ErbB2 mRNA expression over 6, 24 and 48 h incubation. However, the results were also not significantly different (*p* > 0.05).

### 3.4. Detection of Total ErbB1 and ErbB2 Proteins

ErbB1 and ErbB2 proteins were expressed in both Walker 256 (Figure 4a,b) and IEC-6 (Figure 5a,b) cells at all time points but with different staining intensities. Staining can be seen dispersed throughout the cells, with higher intensity at the cytoplasmic membrane of the cells. The nuclei were round and stained blue by DAPI. The results of lapatinib-treated cells were compared with control untreated cells. In Walker 256 breast tumour cells, there were no differences in the ErbB1 (Figure 4a) staining intensity between untreated and lapatinib-treated cells at 6 and 24 h (*p* > 0.05). However, at 48 h incubation, lapatinib-treated cells showed significantly lower ErbB1 staining intensity compared to control untreated and other untreated and lapatinib-treated cells at different incubation period (*p* < 0.0001). Lower cell counts were seen in lapatinib-treated cells at 48 h incubation compared to others (Figure 4a). Similarly, ErbB2 staining intensity in Walker 256 cells incubated with lapatinib for 48 h were the most significantly lower compared to control untreated and other untreated and lapatinib-treated cells at different incubation period (*p* < 0.0001) (Figure 4b). Staining intensities of ErbB1 (Figure 5a) and ErbB2 (Figure 5b) in IEC-6 small intestinal epithelial cells treated with lapatinib were significantly lower than untreated cells at all incubation h (*p* < 0.001–0.0001). Lower cell counts were also seen in lapatinib-treated cells compared to untreated cells (Figure 5a,b). Cell shrinkage was seen in the untreated cells, while lapatinib-treated cells showed slightly increased cell size compared to the untreated cells. Lower cell counts and shrinking cells in control untreated at 48 h may be due to serum-deprivation.

## 4. Discussion

Lapatinib is a dual inhibitor of ErbB1/ErbB2 which is used to treat ErbB2-positive breast cancer. The formation of ErbB1/ErbB2 or ErbB2/ErbB3 heterodimers can enhance the ligand binding, receptor tyrosine phosphorylation and cell proliferation compared to ErbB1 homodimers, thus, lapatinib has superior potency compared to single inhibitors of ErbB1 in inhibiting signal transduction of tumour proliferation and survival pathways [23].

Evaluation of the antiproliferative effect of lapatinib on rat breast tumour and jejunal cell lines indicated that lapatinib showed cytotoxic effects on both cell lines. The cytotoxic effect of lapatinib on Walker 256 rat breast tumour and IEC-6 rat jejunal cell lines was confirmed via XTT assay. However, it was shown that lapatinib is more potent in inhibiting jejunal cell growth compared to breast tumour cell growth. The IC_50_ value of lapatinib on Walker 256 rat breast tumour cell line (8.4 µM) was comparable to the IC_50_ value of lapatinib on human breast tumour cell lines; CAMA-1 (8.3 µM) and MDA-MB-435 (8.5 µM) [12]. While IEC-6 rat jejunal cell line (3.0 µM) showed slightly higher IC_50_ compared to HME human normal mammary (1.34 µM), but lower than HFF human fibroblast (6.45 µM) [13]. The cytotoxic effect of lapatinib on IEC-6 rat jejunal was not able to be compared with either human or rat normal intestinal cells as lapatinib has never been tested on these cell lines. Lapatinib also has never been tested on rat tumour cell lines. It is noted that all of the human cell lines mentioned above possessed moderate or low levels of both ErbB1 and ErbB2 [12,13], although lapatinib showed higher cytotoxic effects on the cell lines. It is important to note that this study provides the first information on the effect of lapatinib on rat breast tumour cells which may contribute to the understanding of its usage in developing animal tumour model to study tyrosine kinase inhibitor-induced diarrhoea.

In FACS analysis, the mechanisms of cell death induced by lapatinib on Walker 256 and IEC-6 cells were evaluated. Cells that were treated with lapatinib at different incubation periods were then sorted and were classified as viable, early apoptosis, late apoptosis and necrosis. Viable cells have an intact membrane, while early apoptotic cells are dying cells in which the plasma membrane remains intact but exposes phosphatidylserine (PS) on the cell surface to mediate its recognition by phagocytes. Early apoptotic cells can become late apoptotic cells, also known as secondary necrotic cells, when the plasma membrane becomes permeabilised [24,25]. Alternatively, direct exposure of healthy viable cells to trauma, such as extreme temperature or, mechanical and chemical insults, can lead to the generation of membrane-permeabilised necrotic cells which are also known as primary necrotic cells [24,25]. Lapatinib was shown to induce early apoptosis in Walker 256 tumour cells at early incubation with lapatinib, however at 48 h incubation, lapatinib induced necrosis in Walker 256 tumour cells. In IEC-6 cells, lapatinib induced late apoptosis at 24 h as well as at 48 h incubation. These findings show that lapatinib causes tumour cell necrosis which explains the drug’s role in tumour inhibition, however, lapatinib-induced late apoptosis in intestinal cells might well be involved in gastrointestinal toxicity which leads to diarrhoea in patients administered with the drug. Nevertheless, further investigation such as cell cycle analysis will further clarify the mechanism of cell death induced by lapatinib on both tumour and normal intestinal cells.

Previous studies have reported that at a very high level of ErbB2 expression, lapatinib sensitivity is increased, while at a lower level of ErbB2, lapatinib sensitivity is decreased. The lowest sensitivity of lapatinib is mostly seen in cell lines with a low level of ErbB2 and nearly undetectable levels of ErbB1 [13]. These findings support the results obtained from this study which showed lower ErbB2 mRNA expression and undetectable ErbB1 mRNA expression in Walker 256 rat breast tumour cells, thus explaining lower sensitivity of lapatinib on this cell line. Although Walker 256 breast tumour cells are less sensitive towards lapatinib because of lower ErbB2 mRNA expression and undetectable ErbB1 mRNA expression, the drug exhibited cytotoxic effect on the cell line, suggesting involvement of other unknown mechanisms which could be an off-target cytotoxic effect [26]. However, from the results it is important to note that it is not clear if the ErbB1 mRNA expression changed when cells were treated with lapatinib. Other possible reason could also be lapatinib inhibition on ErbB2/ErbB3 heterodimers. However, the expression of ErbB3 mRNA was not assessed in this study as lapatinib is known to be targeting ErbB1 and ErbB2. Furthermore, lower ErbB2 mRNA expression was observed in Walker 256 with no significant difference between untreated and lapatinib-treated cells, showing no significant inhibition on ErbB2 mRNA expression. Expression of ErbB3 mRNA as well as other ErbB receptor mRNAs also has never been tested in Walker 256. Thus, further investigation which includes determination of ErbB3 expression is required as other mechanism might contribute to lapatinib cytotoxic properties on Walker 256 rat breast tumour cells. The relatively high ErbB1 and ErbB2 mRNA expression in IEC-6 jejunal cells compared to Walker 256 tumour cells may partially explain differential sensitivity to lapatinib.

Lapatinib inhibited Walker 256 rat breast tumour cell proliferation but with lower cytotoxic effect, which could be due to lower ErbB2 mRNA with ErbB1 mRNA was not able to be detected in the cells. The drug induced necrosis in Walker 256 rat breast tumour cells, explaining the drug’s role in tumour inhibition. In contrast, lapatinib exhibited higher cytotoxic effect on the IEC-6 rat jejunal epithelial cells that could be due to higher levels of ErbB1 and ErbB2 mRNAs in IEC-6 cells. The drug induced IEC-6 rat jejunal cell death via late apoptosis which might cause gastrointestinal toxicity. Previous works have proven lapatinib ability to reduce tyrosine phosphorylation of ErbB1 and ErbB2, and inactivation of Erk1/2 and AKT, the downstream effectors of cell proliferation and cell survival, respectively [10,12,27,28]. However, in this study the downstream mechanism of cell death induced by lapatinib on both Walker 256 rat breast tumour and IEC-6 rat jejunal cells are not known. Further investigations will be able to explain the downstream mechanisms.

The expression and localisation of ErbB1 and ErbB2 in Walker 256 and IEC-6 cells were seen in the immunofluorescence staining. Both ErbB1 and ErbB2 proteins were expressed in both control untreated and lapatinib-treated Walker 256 and IEC-6 cells. Control untreated and lapatinib-treated cells did not show much difference in staining intensity at different incubation h, except at 48 h incubation in which both control untreated and lapatinib-treated showed lower staining intensity as well as lower cell counts. This could be due to prolonged incubation with lapatinib. Previous studies have shown decrease of growth factor expression in terms of relative protein expression following incubation with tyrosine kinase inhibitors [12,29].

Although ErbB1 mRNA expression was undetectable in Walker 256 cells, ErbB1 protein expression were observed via immunofluorescence staining. However, as far as is known, no studies have been conducted to determine the mRNA expression of ErbB1 or other ErbB family members in Walker 256 cells. Thus, further investigations are required to determine the *ErbB* gene expression in Walker 256 cells. Importantly, an understanding of differential expression of lapatinib targets in cancer and normal gastrointestinal tissue may allow identification of diarrhoea intervention targets that could preferentially protect gastrointestinal epithelium. It is important to note that this study provides the first information on the effect of lapatinib on rat breast tumour cells which may contribute to the understanding of its usage in developing animal tumour model to study tyrosine kinase inhibitor-induced diarrhoea.

In summary, lapatinib exhibited cytotoxic properties on ErbB1/ErbB2 expressing cell lines, with jejunal cells being more sensitive to lapatinib compared to tumour cells. Lapatinib induced necrosis in tumour cells, while inducing late apoptosis in jejunal cells. This may explain lapatinib-induced diarrhoea in patients administered with the drug which could be due to apoptosis of small intestinal epithelial cells leading to barrier disruption and consequently diarrhoea. However, much remains to be learned about the molecular mechanisms related to lapatinib’s cytotoxic effect.

## Figures and Tables

**Figure 1 biomedicines-08-00002-f001:**
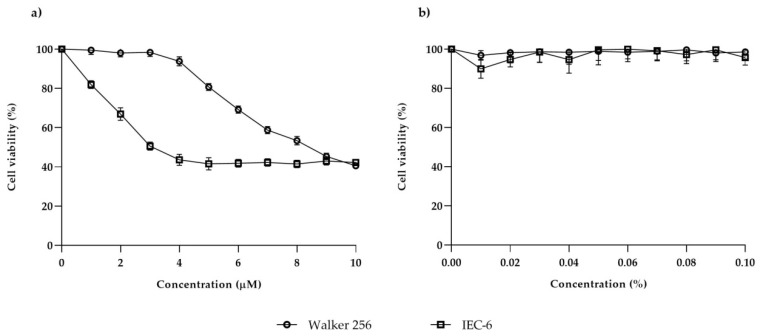
The effect of (**a**) lapatinib and (**b**) dimethyl sulfoxide (DMSO) treatment on Walker 256 and IEC-6 cells as assessed by XTT (2,3-*bis*-(2-methoxy-4-nitro-5-sulfophenyl)-2H-tetrazolium-5-carboxanilide solution) assay (*n* = 4). Data presented as mean ± S.E.M.

**Figure 2 biomedicines-08-00002-f002:**
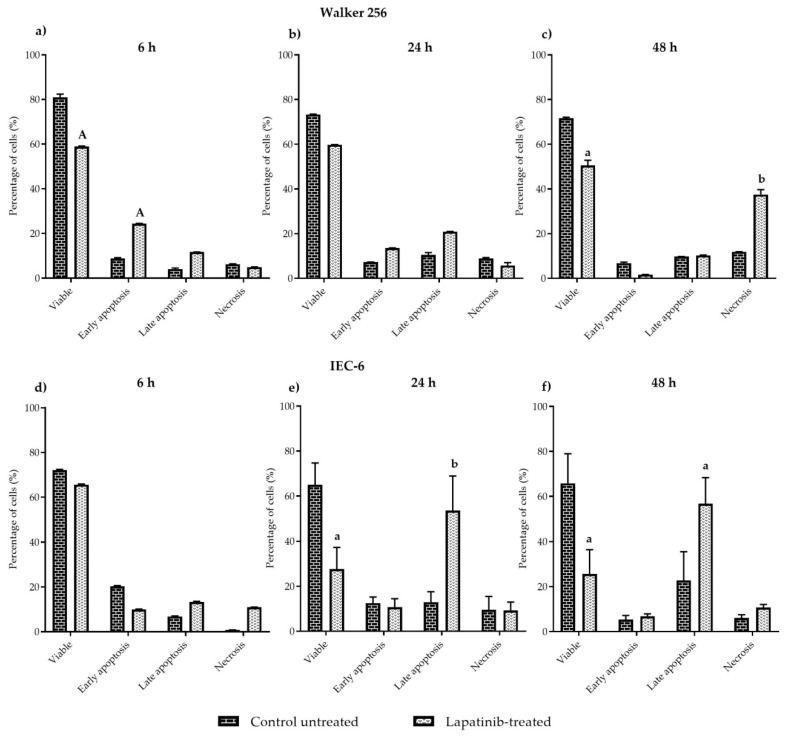
The percentage of viable, early apoptotic, late apoptotic and necrotic cells in lapatinib-treated Walker 256 cells compared to control untreated at (**a**) 6 h (**b**) 24 h (**c**) 48 h incubation and lapatinib-treated IEC-6 cells compared to control untreated at (**d**) 6 h (**e**) 24 h (**f**) 48 h incubation as quantified via FACS analysis. Graph shown for each cell line is representative of experiments conducted. Results shown on the graph are presented as mean ± S.E.M (*n* = 6). Results were compared with control untreated cells at the same incubation time in the same category. Data showing the letters were significantly different at the level of *p* < 0.05. a for *p* < 0.05 compared to control untreated cells, b for *p* < 0.01 compared to control untreated cells, A for *p* < 0.0001 compared to control untreated cells.

**Figure 3 biomedicines-08-00002-f003:**
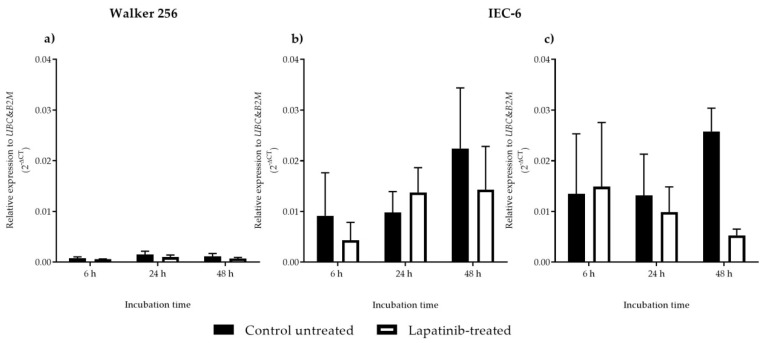
(**a**) *ErbB2* mRNA expression in control untreated and lapatinib-treated Walker 256 cells; (**b**) *ErbB1* and (**c**) *ErbB2* mRNA expression in control untreated and lapatinib-treated IEC-6 cells. Data presented as mean ± S.E.M (*n* = 4).

**Figure 4 biomedicines-08-00002-f004:**
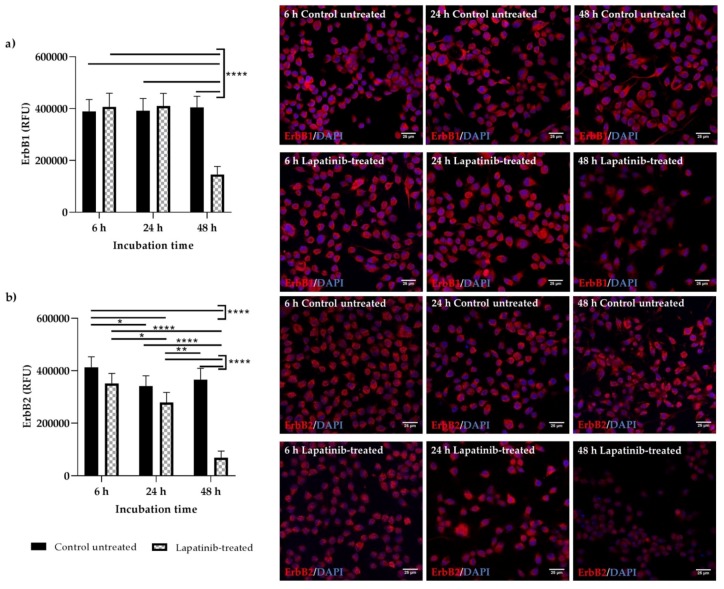
Immunofluorescence staining of ErbB1 and ErbB2 in control untreated and lapatinib-treated Walker 256 rat breast tumour cells at 6, 24 and 48 h incubation. Images shown are overlay images of (**a**) ErbB1 (red)/DAPI (blue) and (**b**) ErbB2 (red)/DAPI (blue). ErbB1 and ErbB2 proteins were expressed in Walker 256 at all time points but with different staining intensities. Corrected total cell fluorescence (CTCF) for ErbB1 and ErbB2 staining intensities were expressed as relative fluorescence units (RFU) as shown on the graphs. Data presented as mean ± S.E.M (*n* = 6). Results were compared between control untreated and lapatinib-treated cells at different incubation h. Significant differences between groups are indicated by bars. * for *p* < 0.05, ** for *p* < 0.01 and **** for *p* < 0.0001. The staining intensities of control untreated and lapatinib-treated cells were also compared with negative controls (1% BSA and Rabbit IgG) which were stained with secondary antibody only (red) (images not shown). Images are of ×60 original magnification (Scale bar: 25 µm).

**Figure 5 biomedicines-08-00002-f005:**
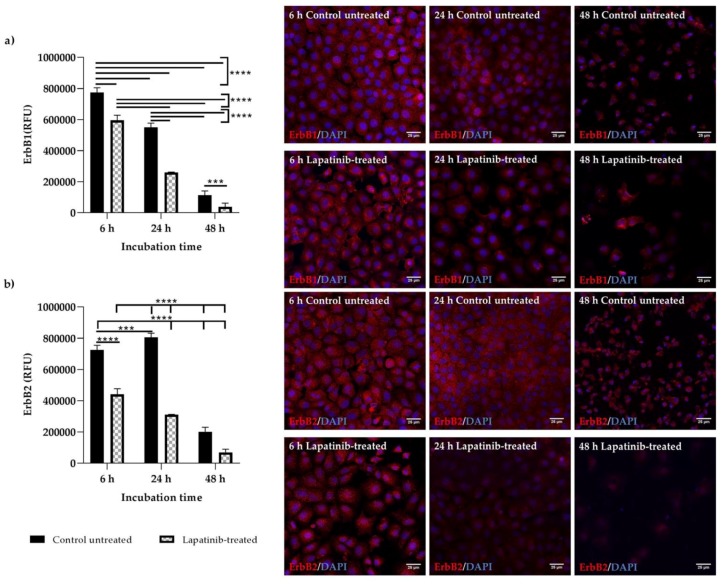
Immunofluorescence staining of ErbB1 and ErbB2 in control untreated and lapatinib-treated IEC-6 rat normal small intestinal epithelial cells at 6, 24 and 48 h incubation. Images shown are overlay images of (**a**) ErbB1 (red)/DAPI (blue) and (**b**) ErbB2 (red)/DAPi (blue). ErbB1 and ErbB2 proteins were expressed in IEC-6 at all time points but with different staining intensities. Corrected total cell fluorescence (CTCF) for ErbB1 and ErbB2 staining intensities were expressed as relative fluorescence units (RFU) as shown on the graphs. Data presented as mean ± S.E.M (*n* = 6). Results were compared between control untreated and lapatinib-treated cells at different incubation h. Significant differences between groups are indicated by bars. *** for *p* < 0.001 and **** for *p* < 0.0001. The staining intensities of control untreated and lapatinib-treated cells were also compared with negative controls (1% BSA and Rabbit IgG) which were stained with secondary antibody only (red) (images not shown). Images are of x60 original magnification (Scale bar:25 µm).

**Table 1 biomedicines-08-00002-t001:** Primer sequences used in real-time polymerase chain reaction analysis.

Gene	Nucleotide Sequence (5′-3′)	Nucleotide Position	Amplicon Length (bp)	Tm (°C)	NCBI Accession No.
*ErbB1*	F:CCCACAGCAAGGCTTCTTCA	3181–3301	119	61; 61	NM_031507.1
	R:CACGGCAGCTCCCATTTCTA				
*ErbB2*	F:AACCTTTCCTTGCTGCTTGA	4021–4201	212	57; 57	NM_017003.2
	R:GTTCCCTCCAGACCTCTTCC				
*UBC*	F:TCGTACCTTTCTCACCACAGTATCTAG	2406–2487	82	58; 56	NM_017314.1
	R:GAAAACTAAGACACCTCCCCATCA				
*B2M*	F:CGAGACCGATGTATATGCTTGC	286–399	114	55; 55	NM_012512.1
	R:GTCCAGATGATTCAGAGCTCCA

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
