# Peer review of "Cytotoxic Effects of the Dual ErbB Tyrosine Kinase Inhibitor, Lapatinib, on Walker 256 Rat Breast Tumour and IEC-6 Rat Normal Small Intestinal Cell Lines"

_biomedicines, 2019, doi:10.3390/biomedicines8010002_

Round 1

Reviewer 1 Report

The manuscript entitled “Cytotoxic effects of the dual ErbB tyrosine kinase inhibitor, lapatinib, on tumour and intestinal cell lines” reports investigations on Lapatinib, a dual ErbB1/ErbB2 tyrosine kinase inhibitor (TKI). This inhibitor is orally administered but induce diarrhoea. The authors show that Lapatinib induce necrosis in tumour cells, while inducing late apoptosis in normal intestinal cell lines, which could be a possible explanation for the appearance of diarrhoea in patients undergoing treatment.

            This is a very systematic study over the cytotoxic effects of inhibitor lapatinib

The article is well written and data is presented in good shape. It could be considered for publication with several minor changes considering in particular several typos and misuse of words

Author Response

Response to Reviewer 1 Comments

Point 1: The manuscript entitled “Cytotoxic effects of the dual ErbB tyrosine kinase inhibitor, lapatinib, on tumour and intestinal cell lines” reports investigations on Lapatinib, a dual ErbB1/ErbB2 tyrosine kinase inhibitor (TKI). This inhibitor is orally administered but induce diarrhoea. The authors show that Lapatinib induce necrosis in tumour cells, while inducing late apoptosis in normal intestinal cell lines, which could be a possible explanation for the appearance of diarrhoea in patients undergoing treatment.

 This is a very systematic study over the cytotoxic effects of inhibitor lapatinib

The article is well written and data is presented in good shape. It could be considered for publication with several minor changes considering in particular several typos and misuse of words

Response 1: Typo error and misuse of words have been corrected throughout the manuscript.

Reviewer 2 Report

The present report by Mohamad Zain et al. characterizes the cytotoxic effects of the ErbB1/2 inhibitor lapatinib on two rat cells lines, breast cancer cells and normal intestinal cells. From their studies, the authors conclude that Lapatinib exhibits cytotoxic effects on both breast tumor cells (necrosis induction) and normal intestinal cells (apoptosis induction), being the latest more sensitive. Apoptosis of intestinal cells might be the underlying cause for lapatinib-induced diarrhea.

It is not clear why the authors used rat cells lines for their studies when, human cell lines, with more relevance for the effects of lapatinib, are available, and studies with the drug have already been performed in these lines. The relevance and translational value of the studies performed is questionable.

Everall, data presentation is not acceptable. Extensive revision is necessary. Some aspects to consider are detailed below.

The title of the report is misleading and must be modified. The species origin of the cells lines tested must be included. The type of tumour (breast) and normal (intestine) cell lines must be clearly specified.  

In all cases, the number of samples studies must be clearly specified, taking into account that replicates must not be considered as a separate case. For instance, an experiment repeated 4 times with 3 replicates in each is not n=12, but n=4 (data on Fig. 1). This must change, in particular, the errors. All data presented must be reviewed taking this into account.

Avoid data duplication. Data in Fig. 2 and Table 2 are the same.

Analysis performed for data presented in Fig. 3 must be reconsidered. Since two factor (treatment and time) are included, a two-way ANOVA must be applied.

It is not clear how immunhistochemical data was analyzed (section 3.4).  Everything seems to be based on the visual comparison of staining intensities among groups. However, the authors talk about comparisons between groups and differences in staining intensities. This is not acceptable, an objective evaluation (cell counting/image analysis) must be applied.

The characteristics of the IEC-6 cell line must be better stated, pointing out that is an epithelial cell line.

Walker 256 cells had undetectable expression of ErbB1 in basal conditions, however, it is not clear if the expression changed when cells were treated with lapatinib. This must be clarified.

It seems that expression of ErbB3 might be important for lapatinib affects (acting on ErbB2/3 heterodimers9. Therefore, at least, expression of ErbB3 should be determined in the cells lines used, with and without treatment.

Author Response

Response to Reviewer 2 Comments

Point 1: It is not clear why the authors used rat cells lines for their studies when, human cell lines, with more relevance for the effects of lapatinib, are available, and studies with the drug have already been performed in these lines. The relevance and translational value of the studies performed is questionable.

Response 1: Changes have been made within lines 77 to 91 to emphasise the reason why rat cell lines were used in this study.  The sentences are now written as follows:

Line 87-91: Understanding the effect of lapatinib on Walker 256 rat breast tumour and IEC-6 rat normal intestinal cells would increase the potential of the cell lines to be used in developing an appropriate rat tumour model to study tyrosine kinase inhibitor-induced gastrointestinal toxicity, in particular lapatinib-induced gastrointestinal toxicity to reflect breast cancer patient setting receiving lapatinib treatment.

The statements have also been written in other lines throughout the manuscript (Line 390-393, Line 456-459). 

Point 2:The title of the report is misleading and must be modified. The species origin of the cells lines tested must be included. The type of tumour (breast) and normal (intestine) cell lines must be clearly specified.

Response 2: Changes have been made to the title by adding the species origin and the type of the cell lines tested.  The title is now written as “Cytotoxic effects of the dual ErbB tyrosine kinase inhibitor, lapatinib, on Walker 256 rat breast tumour and IEC-6 rat normal small intestinal cell lines”.

Point 3:In all cases, the number of samples studies must be clearly specified, taking into account that replicates must not be considered as a separate case. For instance, an experiment repeated 4 times with 3 replicates in each is not n=12, but n=4 (data on Fig. 1). This must change, in particular, the errors. All data presented must be reviewed taking this into account.

Response 3: Changes have been made to Figure 1 (Line 236-237).  Data has been reanalysed with n=4.  Figure 1 caption is now written as follows:

Line 238-240: Figure 1. The effect of a) lapatinib and b) DMSO treatment on Walker 256 and IEC-6 cells as assessed by XTT assay (n=4).  Data presented as mean±S.E.M.

The IC50 values (mean±S.E.M.) mentioned in the text have also been corrected based on the amendments. The statement is now written as follows:

Line 229-231: Lapatinib was found to inhibit 50% of Walker 256 rat breast tumour cell growth at 8.40 ± 0.83 µM, and at 3.00 ± 0.96 µM in the IEC-6 rat jejunum cell line.

Point 4:Avoid data duplication. Data in Fig. 2 and Table 2 are the same.

Response 4: Table 2 has been deleted due to duplication.  Figure 2 has been amended by adding letters to show statistical significant difference between the data (Line 265-266).  Figure 2 caption is now written as follows:

Line 267-275: Figure 2. The percentage of viable, early apoptotic, late apoptotic and necrotic cells in lapatinib-treated Walker 256 cells compared to control untreated at a) 6 hours b) 24 hours c) 48 hours  incubation and lapatinib-treated IEC-6 cells compared to control untreated at d) 6 hours e) 24 hours f) 48 hours incubation as quantified via FACS analysis.  Graph shown for each cell line is representative of experiments conducted. Results shown on the graph are presented as mean ± S.E.M (n=6).  Results were compared with control untreated cells at the same incubation time in the same category. Data showing the letters were significantly different at the level of p<0.05. a for p<0.05 compared to control untreated cells, b for p<0.01 compared to control untreated cells, A for p<0.0001 compared to control untreated cells. 

Due to deletion of Table 2, statement in Section 3.2 has also been changed to quote Figure 2.  Changes are as follows:

Line 242-264: As indicated in the results above, lapatinib was shown to inhibit cell death in both Walker 256 and IEC-6 cells.  Thus, flow cytometry was carried out to evaluate the mechanism of cell death induced by lapatinib.  Percentage of viable, early apoptotic, late apoptotic and necrotic cells in Walker 256 and IEC-6, after treatment with lapatinib at different incubation time were presented in Figure 2a-f.  At 6 hours, lapatinib-treated samples showed a significantly lower number of viable cells (58.99 ± 3.21 %) (p<0.0001) and higher numbers of early apoptotic cells (24.71 ± 1.39 %) (p<0.0001), compared to control untreated (viable cells: 79.97 ± 0.99 %, early apoptotic cells: 7.30 ± 2.51 %) (Figure 2a) (Table 2), as determined by flow cytometry.  However, lapatinib-treated samples did not show any difference in the percentage of viable, early apoptotic, late apoptotic and necrotic cells at 24 hours incubation (Figure 2b) (Table 2) compared to control untreated samples (p>0.05), while at 48 hours incubation, lapatinib-treated samples were shown to have a lower percentage of viable cells (50.70 ± 7.27 %) (p<0.05) and higher percentage of necrotic cells (37.91 ± 7.08 %) (p<0.01), compared to control untreated samples (viable cells: 71.93 ± 6.71 %, necrotic cells: 11.86 ± 5.62 %) (Figure 2c) (Table 2).

As for IEC-6, the results did not show any significant differences in cell viability at 6 hours incubation (p>0.05) (Figure 2d) (Table 2). However, lapatinib-treated samples at 24 hours incubation showed a lower percentage of viable cells (27.72 ± 9.59 %) (p<0.05) and a higher percentage of late apoptotic cells (53.56 ± 15.37 %) (p<0.01) compared to control untreated samples (viable cells: 65.00 ± 9.70 %, late apoptotic cells: 12.91 ± 4.70 %) (Figure 2e) (Table 2).  Similarly, at 48 hours incubation lapatinib-treated samples showed a lower percentage of viable cells (25.68 ± 10.78 %) (p<0.05) and a higher percentage of late apoptotic cells (56.82 ± 11.53 %) (p<0.05) compared to the control untreated samples that exhibited 65.83 ± 13.11 % alive cells and 22.70 ± 12.81 % late apoptotic cells (Figure 2f) (Table 2).

Point 5:Analysis performed for data presented in Fig. 3 must be reconsidered. Since two factors (treatment and time) are included, a two-way ANOVA must be applied.

Response 5: Apologies as it was actually a typo error in Section 2.7 (Statistical analysis).  Data for Figure 3 has been analysed using two-way ANOVA with Sidak’s multiple comparisons test.  Changes have been made to statements in Section 2.7 which are now written as follows:

Line 217-225: Data were presented as mean ± S.E.M.  All analyses were performed using GraphPad Prism version 7.04. Statistical analysis of ErbB2 mRNA expression results in Walker 256 cells were carried out using Kruskal Wallis two-way ANOVA with Sidak’s multiple comparisons test to identify differences between control untreated samples and treated samples at different incubation time.  One-way ANOVA with Tukey’s multiple comparisons test was used to analyse ErbB1 and ErbB2 mRNA expression in IEC-6 cells.  FACS results were analysed using two-way ANOVA with Tukey’s multiple comparisons test to compare means of each group at different incubation time. Statistical significance was accepted as p<0.05.

Point 6:It is not clear how immunohistochemical data was analyzed (section 3.4).  Everything seems to be based on the visual comparison of staining intensities among groups. However, the authors talk about comparisons between groups and differences in staining intensities. This is not acceptable, an objective evaluation (cell counting/image analysis) must be applied.

Response 6: Correction has been made.  Immunohistochemical fluorescence data/images have been analysed using ImageJ software.  The method section has been amended as follows:

Line 210-215: To evaluate the expression of ErbB1 and ErbB2 (positive fluorescein) in control untreated and lapatinib-treated Walker 256 and IEC-6 cells, immunofluorescent images from six random fields were analysed using ImageJ software (http://imagej.nih.gov/ij/; National Institutes of Health, Bethesda, MD, USA).  Corrected total cell fluorescence (CTCF) was calculated using the following formula:

CTCF = Integrated Density – (Area of selected cell x Mean fluorescence of background readings). 

The statistical analysis section has also been amended as follows:

Line 222-224: ErbB1 and ErbB2 immunofluorescence staining intensities were analysed using two-way ANOVA with Tukey’s multiple comparisons test to compare means of each group at different incubation time.

Section 3.4 has been rewritten as follows:

Line 299-320: ErbB1 and ErbB2 proteins were expressed in both Walker 256 (Figure 4a and 4b) and IEC-6 (Figure 5a and 5b) cells at all time points but with different staining intensities.  Staining can be seen dispersed throughout the cells, with higher intensity at the cytoplasmic membrane of the cells.  The nuclei were round and stained blue by DAPI.  The results of lapatinib-treated cells were compared with control untreated cells.  In Walker 256 breast tumour cells, there were no differences in the ErbB1 (Figure 4a) staining intensity between untreated and lapatinib-treated cells at 6 and 24 hours (p>0.05).    However, at 48 hours incubation, lapatinib-treated cells showed significantly lower ErbB1 staining intensity compared to control untreated and other untreated and lapatinib-treated cells at different incubation period (p<0.0001).  Lower cell counts were seen in lapatinib-treated cells at 48 hours incubation compared to others (Figure 4a).  Similarly, ErbB2 staining intensity in Walker 256 cells incubated with lapatinib for 48 hours were the most significantly lower compared to control untreated and other untreated and lapatinib-treated cells at different incubation period (p<0.0001) (Figure 4b). Staining intensities of ErbB1 (Figure 5a) and ErbB2 (Figure 5b) in IEC-6 small intestinal epithelial cells treated with lapatinib were significantly lower than untreated cells at all incubation hours (p<0.001-0.0001).  Lower cell counts were also seen in lapatinib-treated cells compared to untreated cells (Figure 5a and 5b).  Cell shrinkage was seen in the untreated cells, while lapatinib-treated cells showed slightly increased cell size compared to the untreated cells.  Lower cell counts and shrinking cells in control untreated at 48 hours may be due to serum-deprivation.

Figure 4 and Figure 5 captions have been rewritten as follows:

Line 339-350: Figure 4. Immunofluorescence staining of ErbB1 and ErbB2 in control untreated and lapatinib-treated Walker 256 rat breast tumour cells at 6, 24 and 48 hours incubation.  Images shown are overlay images of a) ErbB1 (red)/DAPI (blue) and b) ErbB2 (red)/DAPI (blue).  ErbB1 and ErbB2 proteins were expressed in Walker 256 at all time points but with different staining intensities.  Corrected total cell fluorescence (CTCF) for ErbB1 and ErbB2 staining intensities were expressed as relative fluorescence units (RFU) as shown on the graphs.  Data presented as mean ± S.E.M (n=6).  Results were compared between control untreated and lapatinib-treated cells at different incubation hours.  Significant differences between groups are indicated by bars.  * for p<0.05, ** for p<0.01 and **** for p<0.0001.  The staining intensities of control untreated and lapatinib-treated cells were also compared with negative controls (1% BSA and Rabbit IgG) which were stained with secondary antibody only (red) (images not shown).  Images are of x60 original magnification (Scale bar:25 µm).

Line 357-368: Figure 5. Immunofluorescence staining of ErbB1 and ErbB2 in control untreated and lapatinib-treated IEC-6 rat normal small intestinal epithelial cells at 6, 24 and 48 hours incubation.  Images shown are overlay images of a) ErbB1 (red)/DAPI (blue) and b) ErbB2 (red)/DAPi (blue).  ErbB1 and ErbB2 proteins were expressed in IEC-6 at all time points but with different staining intensities.  Corrected total cell fluorescence (CTCF) for ErbB1 and ErbB2 staining intensities were expressed as relative fluorescence units (RFU) as shown on the graphs.  Data presented as mean ± S.E.M (n=6).  Results were compared between control untreated and lapatinib-treated cells at different incubation hours.  Significant differences between groups are indicated by bars.  *** for p<0.001 and **** for p<0.0001.  The staining intensities of control untreated and lapatinib-treated cells were also compared with negative controls (1% BSA and Rabbit IgG) which were stained with secondary antibody only (red) (images not shown).  Images are of x60 original magnification (Scale bar:25 µm).

Point 7:The characteristics of the IEC-6 cell line must be better stated, pointing out that is an epithelial cell line.

Response 7: The characteristics of IEC-6 cell line as an epithelial cell line has been stated in Line 83 (Introduction section).  The characteristic as an epithelial cell line have also been added in Line 315, 359 and 435. 

Point 8:Walker 256 cells had undetectable expression of ErbB1 in basal conditions, however, it is not clear if the expression changed when cells were treated with lapatinib. This must be clarified.

Response 8: The statement has been added in the text (Line 401-402) to clarify that although Walker 256 cells had undetectable expression of ErbB1, however it is not clear if the expression changed when cells were treated with lapatinib.

Point 9:It seems that expression of ErbB3 might be important for lapatinib effects (acting on ErbB2/3 heterodimers). Therefore, at least, expression of ErbB3 should be determined in the cells lines used, with and without treatment.

Response 9: The authors acknowledge the importance of determining ErbB3 expression and have clarified in the discussion section from Line 421 to 430: Other possible reason could also be lapatinib inhibition on ErbB2/ErbB3 heterodimers.  However, the expression of ErbB3 mRNA was not assessed in this study as lapatinib is known to be targeting ErbB1 and ErbB2.  Furthermore, lower ErbB2 mRNA expression was observed in Walker 256 with no significant difference between untreated and lapatinib-treated cells, showing no significant inhibition on ErbB2 mRNA expression.  Expression of ErbB3 mRNA as well as other ErbB receptor mRNAs also has never been tested in Walker 256.  Thus, further investigation which includes determination of ErbB3 expression is required as other mechanism might contribute to lapatinib cytotoxic properties on Walker 256 rat breast tumour cells.  The relatively high ErbB1 and ErbB2 mRNA expression in IEC-6 jejunal cells compared to Walker 256 tumour cells may partially explain differential sensitivity to lapatinib.

Round 2

Reviewer 1 Report

The revised manuscript is acceptable for purification. The authors have well interpreted the requests.

Reviewer 2 Report

The applied/translational value of the data presented, as it related to the use of rat cells lines vs. human cell lines (that are available) is still a concern.

The report is per se of limitted value within the file.